# Soil Compaction under Different Traction Resistance Conditions—A Case Study in North Italy

Kaihua Liu [1,*], Marco Benetti [1], Marco Sozzi [1], Franco Gasparini [1] and Luigi Sartori [1,2,*]

1 Department of Land, Environment Agriculture and Forestry, University of Padova, 35020 Legnaro, Italy
2 NEOS SRL, Spin-Off of the University of Padova, 35129 Padova, Italy
* Correspondence: kaihua.liu@phd.unipd.it (K.L.); luigi.sartori@unipd.it (L.S.)

**Abstract:** Tractive efficiency is essential in tillage operations to optimise traction performance. In this field experiment, the tractor performance was measured under different traction resistance conditions. This study quantified the soil stress, soil bulk density, soil moisture, soil cone index, soil surface disturbance, rolling resistance and slip rate under different numbers of passes and traction conditions. The actual power used under different soil and traction conditions was collected. Fuel consumption and savings were calculated between uncompacted soil, compacted soil and the permanent traffic lane. The results show that soil stress increases in each location as traction and the number of passes increase. Soil's physical properties increase, such as the soil bulk density, soil cone index and soil surface disturbance. Additionally, the slip rate increases with traction in each soil condition as uncompacted soil, compacted soil and the permanent traffic lane. The results show that the permanent traffic lane has a lower slip rate under different traction conditions than the uncompacted and compacted soil. Furthermore, the permanent traffic lane has less energy consumption with the same traction resistance. The permanent traffic lane saved 25.50%, 29.23% and 42.34% fuel compared to the uncompacted field in 7.85, 14.71 and 24.52 kN traction conditions, respectively. Our results confirm that dynamic factors such as traction and rolling resistance should be considered in soil compaction research rather than static weight only. In practice, the controlled traffic farming (CTF) system or driving the tractor more frequently on the permanent traffic lane should be considered to improve working efficiency and reduce energy consumption.

**Keywords:** soil compaction; traction; slip rate; rolling resistance



## 1. Introduction

Soil compaction has become an increasingly severe problem in the past few decades, with the size and weight of agricultural machines increasing worldwide [1]. Researchers in different countries have studied the negative effects caused by soil compaction [2–5]. The previous study shows that soil compaction could increase soil bulk density, soil cone index and soil shear strength [6–8]. In addition, soil compaction causes the impeding of root exploration, reduced crop yield and increased energy requirements in all field operations [9–13].

Soil compaction is defined by the Environmental Assessment of Soil for Monitoring (ENVASSO) as "The densification and distortion of soil by which total and air-filled porosity are reduced, causing a deterioration or loss of one or more soil functions" [14]. It can destroy or over-squeeze soil aggregates to form a damaged layer of soil structure during the first pass [15–18]. After the compaction, the soil particles inside the damaged layer are rearranged under external force, decreasing soil volume and increasing soil bulk density [19,20].

As for soil compaction measurement, different methods may have different advantages and disadvantages. The load cell [21,22] measures the soil stress generally in the vertical direction of the target position but provides only one stress component. Stress-state

transducer [23,24] contains load cells in six different directions. However, ensuring good probe contact with the soil can be difficult due to the complex probe geometry of the sensor. The fluid-filled flexible pressure probe [25–27] is directly related to the mean normal stress, which is simple and quick to install and has good probe contact with the soil. However, the pressure reading is still a function of the Poisson's ratio of the soil, which may change during the compaction process.

The risk of soil compaction can be reduced by (I) avoiding entering the field in wet soil conditions [4,28–30], (II) using tyres with a larger contact area and low inflation pressure [31–33] and (III) introducing the controlled traffic farming (CTF) system [11,34–38].

Tyre inflation pressure and wheel load are also well-known key drivers of compaction. Many researchers work on the soil stress distribution underneath the soil caused by different properties of soil, tyre and machine weight conditions [33,39–42]. However, most of them focused on static wheel load rather than dynamic wheel load, which is the actual weight that compacts the soil. Dynamic wheel loads are higher than static wheel loads. For example, the wheel load could increase by 25% of the static wheel load during plough [43]. The extra weight is caused by the traction required to carry out the plough operation and the weight redistribution from the front to the tractor's rear axle. The increase in rear wheel load during conventional tillage given in the guidelines of the German Engineers' Association [44] is even higher (up to 45%), which causes much more compaction than the static wheel load.

Drawbar pull, travel reduction (slip) and rolling resistance are the three main criteria describing off-road vehicles' traction behaviour. The lugs on tractor wheels tend to penetrate deep into the soil layer in terms of working on the soft ground that characterises almost all agricultural operations [45–47]. The tractor lugs compress soil horizontally, opposite to the tractor movement when they are dug into the soil. As a result, the speed at which the tractor moves decreases. This loss of relative speed of the tractor is estimated as the slip coefficient [46,48–50]. Maximum traction can go into slippage, increasing the soil structure damage [46,50–53].

Controlled traffic farming (CTF) is a mechanisation system in which all machinery has the same (or modular) working and track width so that field traffic can be confined to the least possible area of permanent traffic lanes [54,55]. The CTF is one of the solutions for reducing soil compaction, ensuring that the tractor travels on the traffic lane, which has a more solid soil structure than conventional agriculture. In addition, the permanence of the non-pavement surface facilitates the maintenance of softer soil conditions, thus reducing resistance and energy requirements during field operation [11,56,57]. The reduced energy consumption of CTF systems is also attributable to the lower rolling resistance and slippage of tyres on permanent traffic lanes [11,50,58]. Tractor slip rate and traction efficiency are critical parameters for farm operations. Although many studies have demonstrated that the CTF system has lower slippage and higher traction efficiency than conventional agriculture, these two parameters have been less studied in conventional fields under different compaction times and traction resistance conditions. The effect of different levels of soil compaction on the tractor's working efficiency under conventional farming is unclear. Most of the previous research was conducted to increase traction resistance by increasing the weight of the load [59–65]. However, this research approach cannot focus on the effect of traction resistance on soil compaction because of the machine's added weight. Thus, in this experiment, the method of increasing the traction resistance without adding weight was implemented.

The aim of our study was to test the effects of different traction conditions on soil bulk density, soil moisture, soil cone index, soil surface disturbance, slip rate and tractor working efficiency. In this experiment, a method to increase the traction force is proposed. A method was set to increase the traction force independently rather than increasing the weight simultaneously, as in the previous research [65]. First, we tried to determine if the wheel slippage rate would be less under more wheel passes in this experiment. Then, we

focused on whether the soil structure damage would be more significant at higher traction forces and a greater number of wheel passes.

## 2. Materials and Methods

The field experiment lasted from March to May 2022. The experiment field used for this study is located at the experimental farm of Padova University (Veneto, Italy). The area of the field is 1.87 hectares. The slope of the field is less than 1°, measured by Google Earth Pro [66]. Temperatures rise from January (min average: −1.5 °C) to July (max average: 27.2 °C). The sub-humid climate receives about 850 mm of rainfall annually, with the highest average rainfall in June (100 mm) and October (90 mm). The lowest averages happen in winter (50–60 mm). Soil moisture was 23.01%, 23.54% and 27.01% at 20 cm, 40 cm and 60 cm depth when the experiment started. No rainfall occurred during the experiment. The soil texture of the experiment field is clay loam [67]. The sand–silt–clay content of the soils used for testing was 33.8%, 37.0% and 29.2%, respectively. The organic matter content of the topsoil (0–30 cm) was 1.81%, referring to another field also located at the experimental farm of Padova University [67] (straight line distance not exceeding 200 m between two fields). The field was deep ploughing (0–50 cm) after the 2021 crop season, which has a partly bare surface. There were no other field management practices until the start of the field experiment.

The field experiment was preceded in the field as shown in Figure 1. First, slip rates and rolling resistance under different traction and soil conditions were performed in the left part of the field. Then, the field's right area was set into small plots located after the 40 m long area for the stable driving speed of the tractors (3.3 km/h) during the test, as shown in Figure 1b. The blue box area is the place where we collected the data. The 6 m area between each plot was designed for the tractors turning after each round. The detailed data collection procedure for the experiments is listed below.

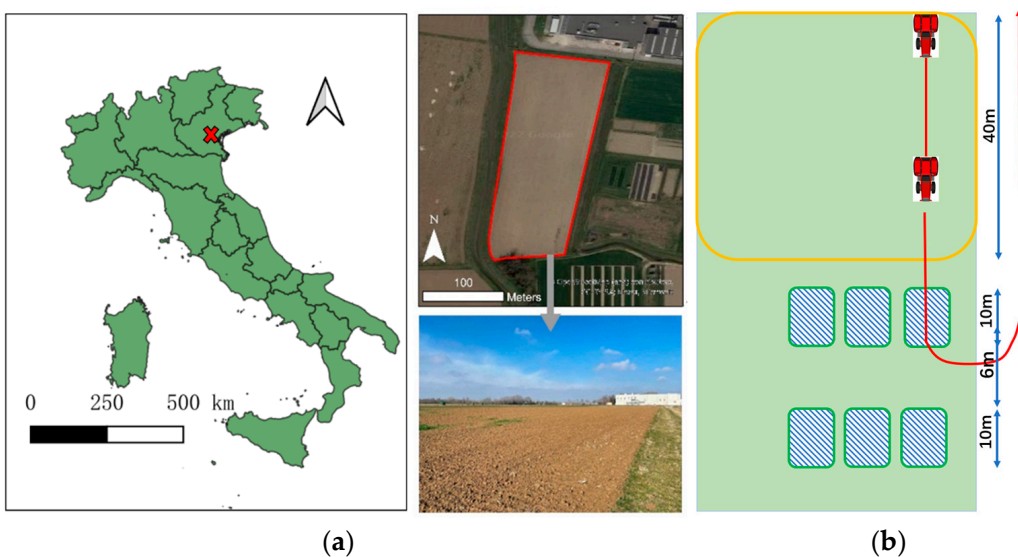

(**a**)　　　　　　　　　　　　　　　　　(**b**)

**Figure 1.** Experiment field in Legnaro, North Italy (**a**), and the experimental procedure (**b**).

Two tractors were used in the experiment. The front tractor was used to compact the soil in different traction conditions and number of passes. The rear tractor was designed to use the hydrostatic transmission system to adjust the braking force.

The front tractor was a Fiat 680 (CNH Industrial N.V., Amsterdam, The Netherlands), as shown in Figure 2. The machine's weight was increased by adding additional counterweights to the back of the front tractor to increase the soil pressure on the tyre and soil contact surfaces for better stress detection and analysis.

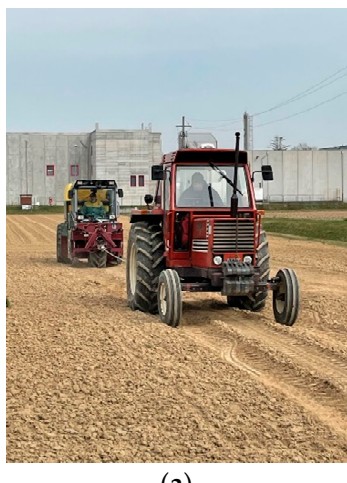

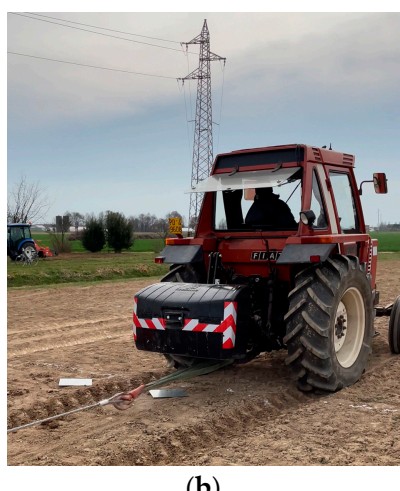

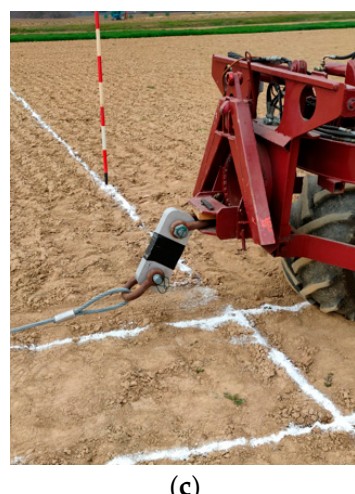

(**a**)                                          (**b**)                                          (**c**)

**Figure 2.** Two tractors used in the field experiment (**a**), counterweight attached in the back of the front tractor (**b**) and dynamometer between the tractors (**c**).

A 16 m wire rope connected the two tractors to ensure enough space to prevent the rear tractor from compacting the experimental field. A dynamometer (TZR 20 t, Yale Industrial Products GmbH, Wuppertal, Germany) was placed on the front of the rear tractor, which measures and transmits the real-time traction data to the monitor of the rear tractor, as shown in Figure 2c. This real-time data transmission guides the tractor driver to adjust the forward speed to keep the braking load (traction) constant at a pre-set value.

The specific parameters of the tractor are shown in Table 1. A tractor prototype three-wheeler was used at the back, equipped with a hydrostatic gearbox, which made it possible to change the traction resistance during the experiment. As a rear tractor, the high responsiveness of the hydrostatic gearbox was used to regulate the forward speed and braking capacity to generate different amounts of traction.

**Table 1.** Technical data of the tractors used in the experiment.

| Name | Unit | Model |
|---|---|---|
| Tractor Model | | Fiat 680 |
| Total mass | kg | 4310 |
| Front axle | kg | 780 |
| Rear axle | kg | 3530 |
| Rear tyre | Kleber traker | 420/85R30 |
| Front tyre | Vredestein multirill | 7.50–16 |
| Front tyre inflation pressure | bar | 1.7 |
| Rear tyre inflation pressure | bar | 1.45 |

The actual speed was recorded from the GPS. The real-time kinematic positioning (RTK) system from Trimble was equipped on the front tractor, which was used to record the track with high accuracy during the experiment. For theoretical speeds, the sensor was placed on the tractor power take-off unit (PTO) to detect the number of revolutions. We calculated the theoretical speed of the tractor by detecting the PTO rpm and measuring the tractor's fixed gear ratio.

*2.1. Mean Normal Stress*

Normal stresses underneath the soil were measured using the Bolling probe [25,27] in the field experiment. The probe is deformable and cylindrical, and could sense the mean radial stress experienced. For the installation of the Bolling probe, the drill and reamer were inserted into the soil at a specific angle on the side of the probes by using a special steel frame which could ensure the angle consistency during the installation of the drill,

reamer and Bolling probe. After the completion of reaming, the probe was inserted into the soil and tested for good contact with the soil to ensure accurate data collection.

In this experiment, the mean normal stress of the soil was measured in the vertical direction and also in the lateral direction. The soil mean normal stress in the vertical direction can be applied to the depth of the subsoil (0–100 cm) [68]. However, the lateral compaction affects shallower soil [32], also verified by the simulation results. Considering the amount of pressure that can occur at different positions, three Bolling probes were used in the vertical direction (0–60 cm) and two in the lateral position (0–40 cm) in this measurement.

Three probes were installed in the centre of the track at 20, 40 and 60 cm depth to measure the soil mean normal stress in the vertical direction. Two probes were installed in the track edge at 20 and 40 cm depth to measure the soil mean normal stress in the lateral direction. There were five groups overall: 20 cm depth in vertical (20 V), 40 cm depth in vertical (40 V), 60 cm depth in vertical (60 V), 20 cm depth in lateral (20 L) and 40 cm depth in lateral (40 L). The width between the centre and the lateral is 25 cm. The stress data under different depths were collected after each time compaction (9 times in total). The probes were inserted into the soil from the edge side of the wheel rut, as shown in Figure 3.

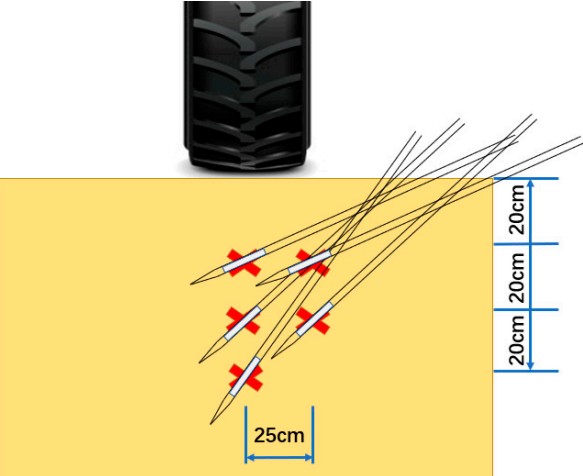

**Figure 3.** Installation position of Bolling probe to measure the mean normal stress in vertical (3 probes) and lateral (2 probes) directions. The three probes in the vertical direction and the two probes in the lateral direction were staggered during the installation, with 25 cm between two groups.

The mean normal stress ($\sigma_m$) was calculated [25,27] to compare the results of different traction conditions and number of passes. The following is the calculation formula of the mean normal stress.

$$\sigma_m = \frac{1+v}{3(1-v)} p_i$$

where $p_i$ measures stress from the Bolling probe, and $v$ is the Poisson ratio in the soil matrix. The value of the Poisson ratio was considered within 0.2–0.45 [40,69–72]. We set the Poisson ratio as 0.3 in our study, considering the results of other studies [25,70].

The mean normal stress collected from the Bolling probe was compared with the simulation results produced by Terranimo [73,74]. Terranimo is a computer model that predicts the risk of soil compaction by farm machinery [75,76]. It includes two inputs (machinery and soil) and two outputs (stresses in the tyre–soil interface and stresses transmitted to the soil profile). An example of the simulation results of the soil stress of the rear tyre is shown in Figure 4. The simulation results and the data collected during the field experiment are given in Section 3. It is worth noting that the soil depths on the y-axis are negative numbers generated automatically by the system. However, in analysing this graph, we default to this problem, as in the graphs in other studies, there is no negative sign in the process of indicating depth [32,65,77].

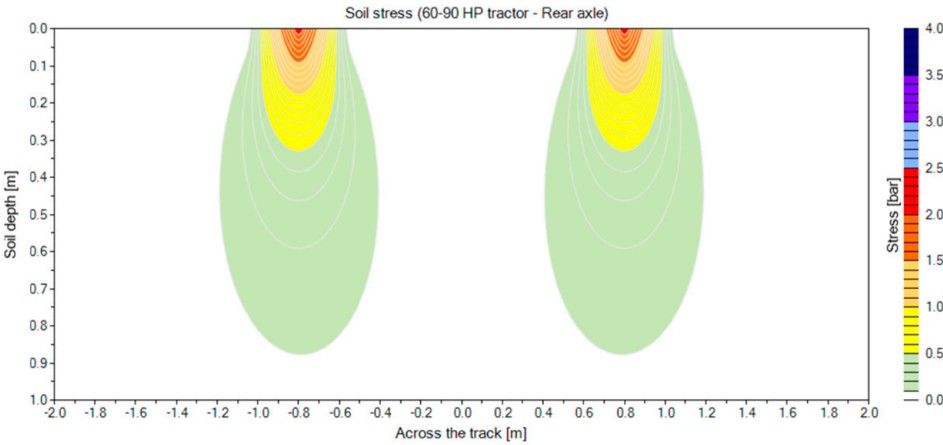

**Figure 4.** Soil stress simulation results of the rear tyre using Terranimo.

### 2.2. Soil Bulk Density and Soil Moisture

The soil bulk density and soil moisture were collected and calculated through the experiment of uncompacted field conditions (0 pass) and each traction condition after 1, 4 and 9 passes. Soil bulk density was collected by the soil sampler (Eijkelkamp, EM Giesbek, The Netherlands). Three groups of soil samples were collected in the vertical direction at 20, 40 and 60 cm depth. Two groups of soil samples were taken on the lateral side of the tyre at 20 and 40 cm depth as shown in Figure 5. There are five groups in total: 20 cm depth in vertical (20 V), 40 cm depth in vertical (40 V), 60 cm depth in vertical (60 V), 20 cm depth in lateral (20 L) and 40 cm depth in lateral (40 L), which are the same as for the soil cone index. Each point was repeated three times. In total, 240 soil samples were collected in the experiment.

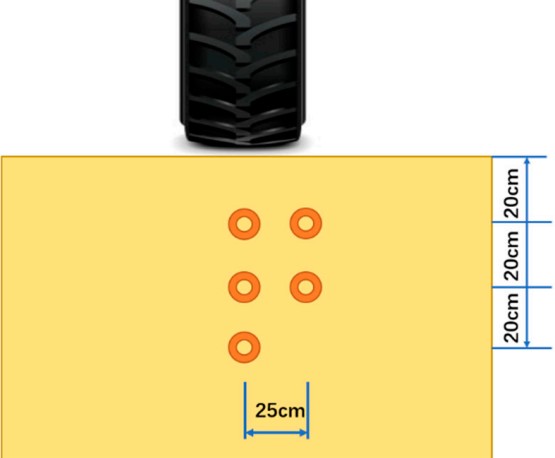

**Figure 5.** The locations of the soil sample collection to measure the soil bulk density in vertical (3 locations) and lateral (2 locations) directions.

### 2.3. Soil Cone Index

Penetration resistance of the soil under different treatments was measured with a penetrometer named Penetrologger (Eijkelkamp, Geesbek, The Netherlands). The probe has a 2 cm$^2$ needle with a 30° cone (standard ASAE S3133 FEB04). Each data group contains a 0–70 cm depth soil cone index collected from the centre line to 50 cm width lateral (5 cm each, 11 points in 50 cm width), as shown in Figure 6. In order to measure the soil cone index from 0 to 70 cm, a hydraulic system was used during the data collection. The penetrometer was mounted on a designed iron frame fixed to the hydraulic piston. The iron frame allows the penetrometer to change the location of measurement horizontally.

The hydraulic piston driven by the tractor allows the uniform insertion speed during the measurement. The soil cone index was recorded in every centimetre of each insert. Each data group was collected before the first pass, and after 1, 4 and 9 passes. Each point in each condition was repeated three times.

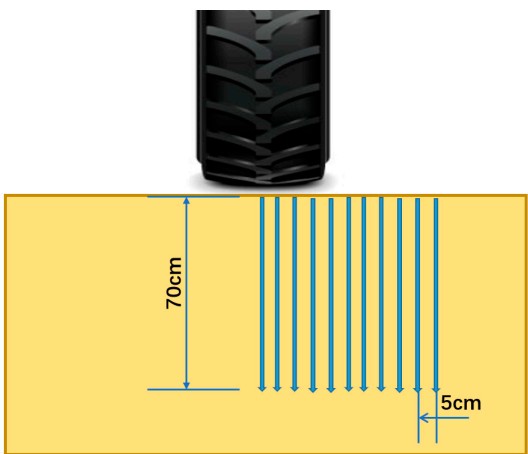

**Figure 6.** Soil cone index measurement using the penetrometer at 0–70 cm depth vertical and 5 cm each horizontal.

The soil cone index measurements were used to analyse the following method [6,78]. The four parts of the results were collected and calculated as (1) the range of the nose zone, (2) the depth of the max cone index, (3) the max cone index and (4) the average cone index from 0 to 40 cm. The nose zone in the soil cone index profile was assumed to be the peak in the profile, which starts and ends with the same soil cone index value, as shown in Figure 7.

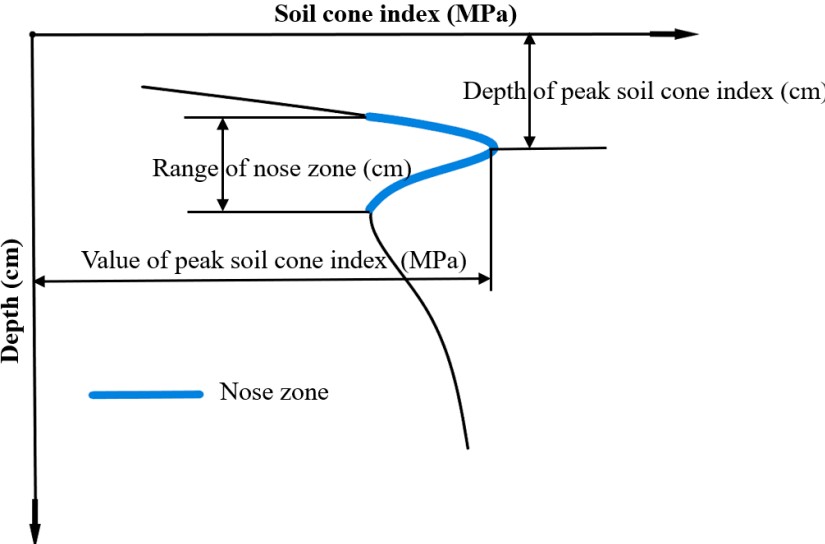

**Figure 7.** Schematic view of cone index (CI)-related traits.

The soil deformation was measured during the experiment to study the soil surface deformation in different traction conditions and number of passes. A steel flame combined with a laser rangefinder (Disto Pro, Leica Geosystems AG, Balagah, Switzerland) measured the distance between the flame and the soil surface horizontally every 2 cm, as shown in Figure 8. Data were collected before the first pass, and after 1, 4 and 9 passes. The data collection was repeated three times.

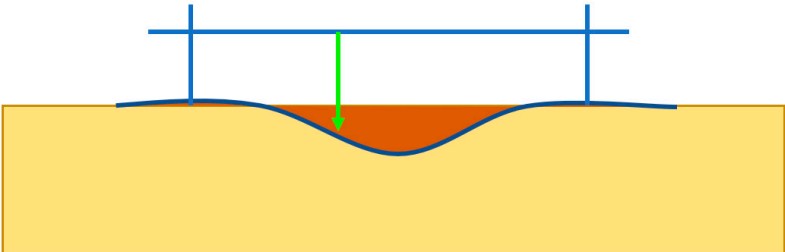

**Figure 8.** Profile meter measurement using a laser rangefinder to calculate the value difference between the compacted and undisturbed soil surface.

*2.4. Slip Rate*

Wheel slip is more likely to cause soil compaction than additional wheel loading, especially for heavier tractors [79]. When the tractor's rear wheel slip rate increases, the maximum shear contact stress rises sharply [80]. This test aimed to assess if and how traction performance varied with the number of passes the tractor made. The slip was calculated as:

$$S = \frac{(d_2 - d_1/4)}{d_2} * 100$$

The $d_1$ is the actual distance after four turns of rear wheels. $d_2$ is the theoretical moving distance during the four-wheel turns, and $S$ represents the slip rate.

Rolling resistance was calculated first by pulling the unload front tractor, simulating nine passes on a predetermined track and using the dynamometer.

In addition, the slippage of the tractor used in the experiment in different traction conditions was tested and measured: 0, 800, 1500 and 2500 kg, which were 0, 7845, 14,710 and 24,517 N, respectively. Using the data collected by the dynamometer, the traction loads were obtained for the traction in four levels. The coefficient of adhesion (ka) was then calculated using the collected rolling resistance and the adherent weight of the machine on the driving axle [81].

$$k_a = c_a / G_a$$

where $c_a$ is the rolling resistance, and $G_a$ is the loading weight.

The machine's inherent losses were subtracted from the power generated by the engine to calculate the actual power used for productive work. The 7 kW and 4 kW power losses were considered while calculating the actual power used for the work because of the transmission and hydraulic system based on the previous study [81]. To estimate the power used during the tractor moving, the formula for the determination of the rolling power ($P_r$) expressed in kW is as follows:

$$P_r = R_r * Va * 10^{-3}$$

where $R_r$ is torque (N), and Va is the rolling speed (m/s). The same formula was used to estimate the useful power (Pu) for each level of traction, considering the previously calculated traction loads. Based on the power used and the type of use, the specific diesel fuel consumption of 260 g/kWh was considered according to data in the literature [81]. Through the relationship between specific consumption and useful power, the fuel consumption in kg/h has been estimated for each level of traction.

The fuel consumption in different traction and soil conditions was calculated. In addition, the diesel saving was calculated to assess diesel fuel consumption in varied soil conditions such as uncompacted soil, compacted soil and field edges (permanent traffic lanes). Finally, the combustion of one litre of diesel fuel produces 2.67 kg of $CO_2$ [82], and the carbon dioxide emissions were estimated in the various simulations. In the experiment, we assume that the density of diesel fuel is 0.85 kg/$dm^3$.

*2.5. Statistics*

In each type of result, the arithmetic mean value of mean normal stress, soil bulk density, soil moisture, soil cone index and soil disturbance were calculated for each position (centre 20, 40 and 60 cm depth, lateral 20 and 40 cm) with a different number of passes and traction conditions as 0, 7.85, 14.71 and 24.52 kN. The three resulting values for each position of each treatment were considered replicates. Statistical analyses of results were undertaken with SPSS [83]. The analysis of variance (ANOVA) was used to compare means with a probability level of 5%.

## 3. Results

*3.1. Stress under the Soil Using the Bolling Probe Sensor*

In this study, a method to incorporate traction and rolling resistance into soil pressure simulations compared with the collected results was implemented. As discussed in the slip rate test section, the rolling resistance calculated is 273 ($\pm$4.79) kg as 2677 ($\pm$46.97) N of the tractor in the experiment. Then, 0 kg, 800 kg, 1500 kg and 2500 kg, which were 0, 7845, 14,710 and 24,517 N, were considered as the extra traction compared to the rolling resistance of the tractor. Therefore, these two parts consist of the net traction, which was considered the additional horizontal pressure with the weight of the tractor. In Terranimo, a slightly higher additional stress was calculated for the rear axle in each simulation. For example, an additional 1000 kg was considered in the simulated rear wheel pressure conditions by simulating 100 investigated tractors at a support load of 3000 kg, which means the total weight to input into the model during the simulation is 4000 kg rather than the actual weight (3000 kg) [73]. The weight was overestimated to avoid underestimating the load and taking into account the additional load transfer effect due to the rolling resistance of the trailer [73]. However, previous studies have not tested the true value of rolling resistance in field tests. In our current study, the tractor gravity, rolling resistance in the vertical direction and traction resistance in the horizontal direction were considered. The values were put into the Terranimo system in different traction conditions to calculate the sum of squares in both directions.

The traction force ($\sigma$) was calculated by calculating the arithmetic sum of squares of the vertical load (W) and the net traction (NT), as shown in Figure 9. The vertical load of the rear tyre was settled during the experiment. The net traction changed between four traction conditions. The net traction was considered located in the rear tyres only because of the two-wheel drive system of the Fiat 680. The traction force ($\sigma$) in different conditions was input into the Terranimo system, and the simulation results were obtained at different locations and depths. After the traction force was calculated, we compared the soil stress data collected from the field experiment with the simulation results, as shown in Figure 10.

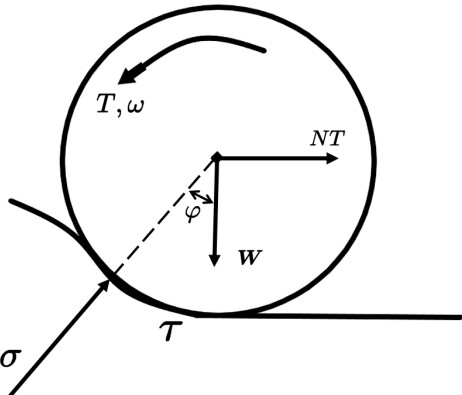

**Figure 9.** Schematic of soil–tyre interaction. Total driving torque on the wheel (T), the angular velocity of the wheel ($\omega$), net traction (NT), the angle between normal stress and the vertical ($\varphi$), vertical load (W), normal stress ($\sigma$) and shear stress ($\tau$).

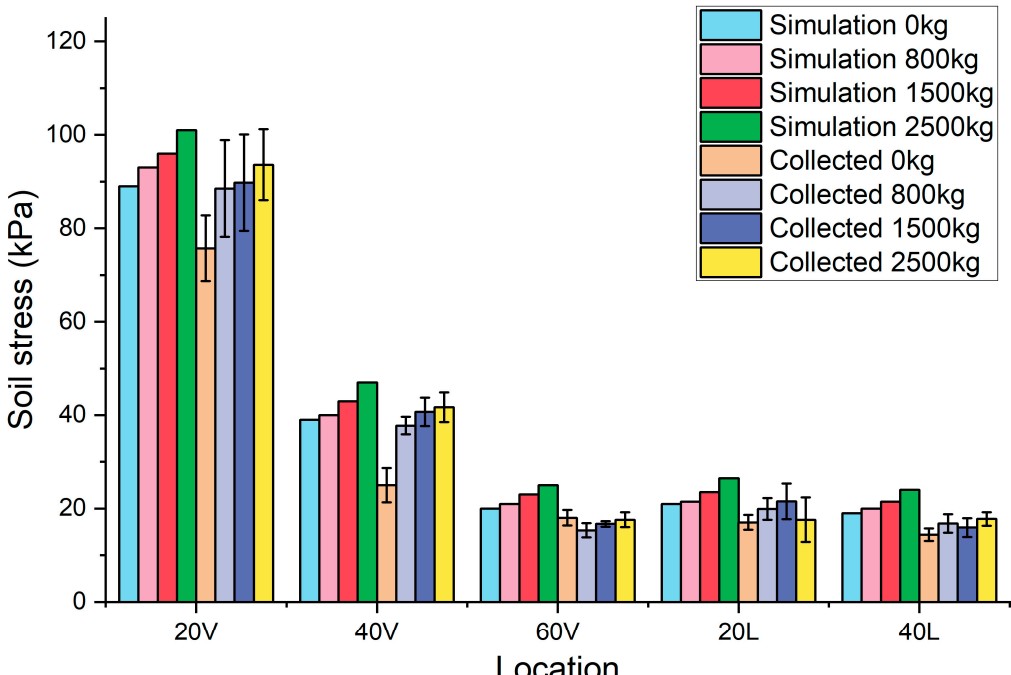

**Figure 10.** Simulated and collected soil stress comparison.

Simulated results and collected data are shown in Figure 10. There are five groups of bars: 20 cm depth in vertical (20 V), 40 cm depth in vertical (40 V), 60 cm depth in vertical (60 V), 20 cm depth in lateral (20 L) and 40 cm depth in lateral (40 L). The result of each bar was made by the average value of one to nine passes in each depth and position. The simulated results have a higher value than the collected data in each depth and direction. The error bars are made by the standard deviation of each position of collected results. The simulation soil stress increases as the traction increases, and the collected data show the same trend in most positions. However, the vertical 60 cm in 0 kg traction shows a higher value than the 800 kg and 1500 kg. The result of the simulation grows uniformly with the increasing traction. The collected results in the vertical 60 cm, lateral 20 cm and lateral 40 cm directions show that the increasing trend with the increasing traction is not distinct compared to the collected results for vertical 20 cm and 40 cm. The subsoil spatial variability could cause this irregular variation in these positions. Furthermore, the data acquisition was likely unsatisfactory in the low-value condition. The probe may have had insufficient contact with the soil in low-pressure conditions compared to high-pressure locations.

### 3.2. Soil Bulk Density and Soil Moisture

As shown in Figure 11, the 60 cm depth had the highest soil moisture in the five positions among the four kinds of traction conditions. In general, the soil moisture slightly increases after compaction. The ANOVA test showed no significant difference in the soil moisture based on the different traction conditions, number of passes and position, except for the moisture in lateral 20 cm in the 1500 kg traction condition. However, there is a slight difference in different traction conditions and the soil moisture in the experiment. The mean difference in soil moisture content is within 1.5%, considering random errors in the data collection process. Studying this experiment with other parameters is recommended for more in-depth analyses and conclusions.

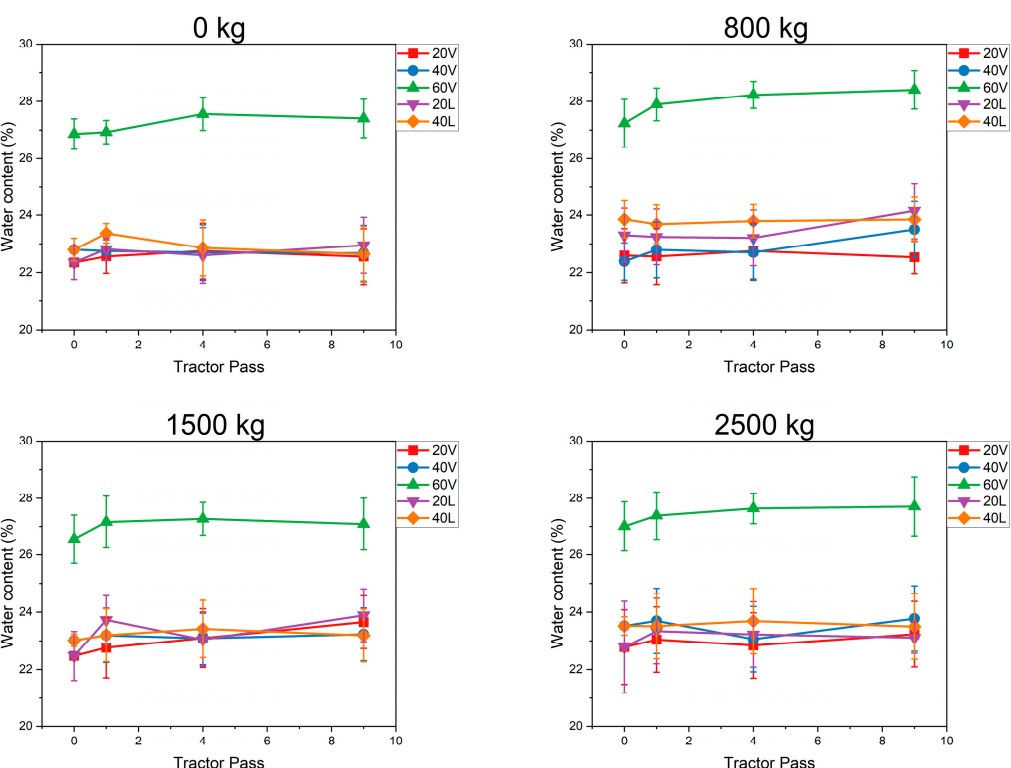

**Figure 11.** Soil moisture content in different traction conditions.

The soil bulk density data in different traction conditions and the number of passes are shown in Figure 12. The results indicate that the soil bulk density increases as the number of passes increases. However, there is no significant difference between different traction conditions of the bulk density, except for 2500 kg traction compared to the other three. Bulk density in vertical and lateral 20 cm positions increased with the number of passes and reached a similar value of 1.41 ($\pm$0.028) g/cm$^3$. As for the 40 cm depth, the bulk density in vertical and lateral directions reached the same range as 1.47 ($\pm$0.020) g/cm$^3$ after nine passes in 0 kg, 800 kg and 1500 kg conditions. However, the vertical 40 cm (1.50 $\pm$ 0.067 g/cm$^3$) shows a higher bulk density value than the lateral 40 cm (1.43 $\pm$ 0.060 g/cm$^3$). The soil bulk density in the vertical 60 cm condition increased with the number of passes in the four traction conditions. After nine passes in four traction conditions, the average bulk density at 60 cm depth is 1.53 $\pm$ 0.005 g/cm$^3$.

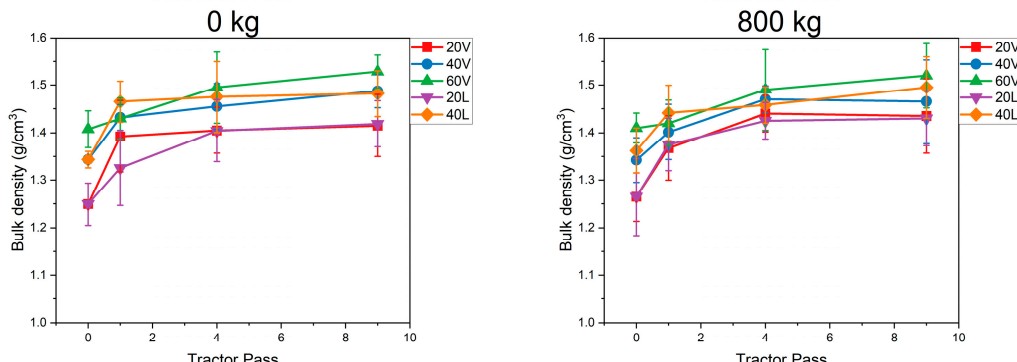

**Figure 12.** *Cont.*

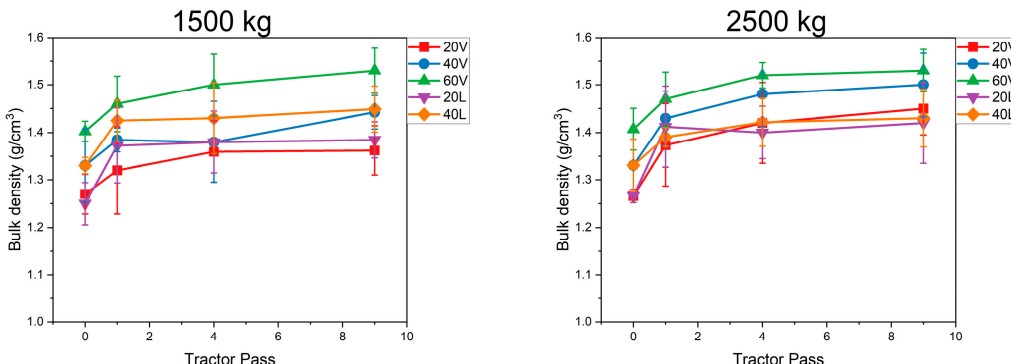

**Figure 12.** Soil bulk density in different traction conditions.

### 3.3. Soil Cone Index

The soil cone index data before the first pass and after one, four and nine passes are shown in Figure 13. The first figure is the average of 0–50 cm data (11 points in total), and the other three are the 0–10 cm data (three points in total in the centre of the rear tyre). All four traction conditions maintained a similar trend for the soil cone index before the first pass. However, one location below 40 cm had a lower soil cone index than the other three (here, we used the word "location" because the four areas were not compacted yet). As for the soil cone index after one, four and nine passes, the peak value increased with traction. It is worth mentioning that the compaction caused the subsidence of the soil surface, so the value begins below the 0 cm depth. However, the peak value of the soil cone index does not show a significant difference under different traction conditions located around 10 to 30 cm depth. The result shows that the main change in the soil cone index happened in the 0–40 cm area because the field's hardpan exists at 40–45 cm depth. Hardpans (plough pans) are formed by years of deep ploughing at the same depth, which stop the compaction at a deeper depth [84].

The soil cone index data of the different traction conditions with zero, one, four and nine passes are shown in Figure 14. In general, each figure shows that the soil cone index increased after compaction in each traction condition. The peak soil cone index increased with the number of passes. However, the soil cone index with different traction conditions and the same number of passes did not show a significant difference. The main change in the soil cone index happened at a depth from 0 to 40, which is the upper part of the hardpan. The soil cone index before the first pass shows a slight difference within four fields. For example, the field with 2500 kg traction conditions shows a higher soil cone index between 10 and 20 cm depth than the other fields.

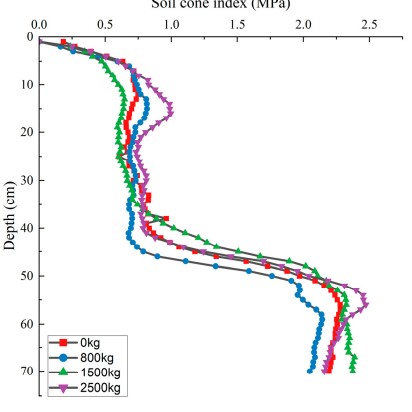

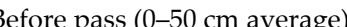

Before pass (0–50 cm average)

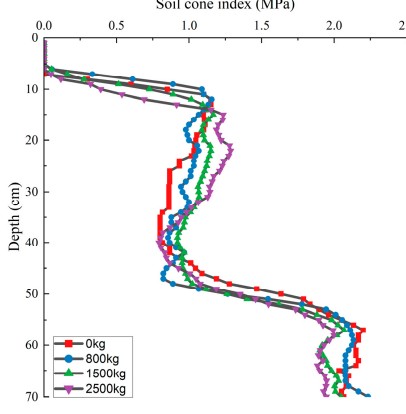

After 1 pass (0–10 cm average)

**Figure 13.** *Cont.*

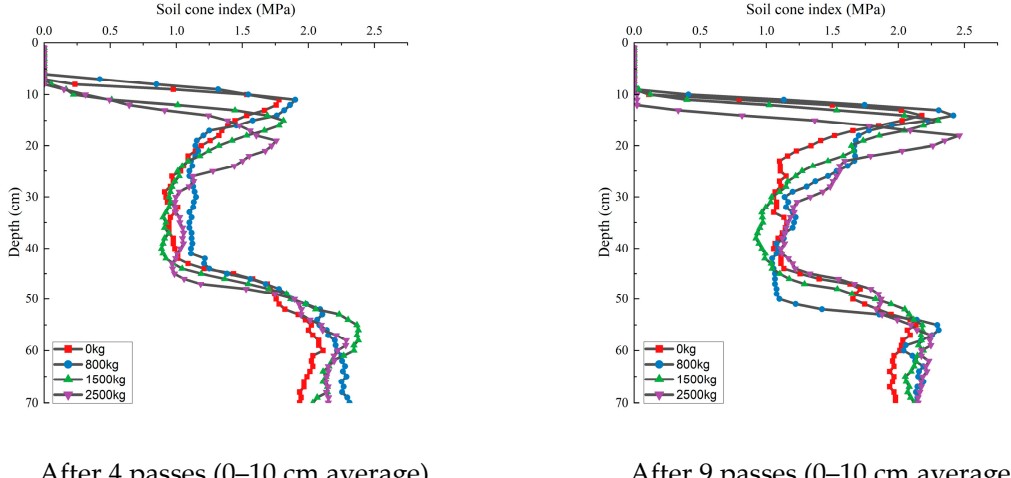

After 4 passes (0–10 cm average)    After 9 passes (0–10 cm average)

**Figure 13.** Soil cone index with different numbers of passes in different traction conditions.

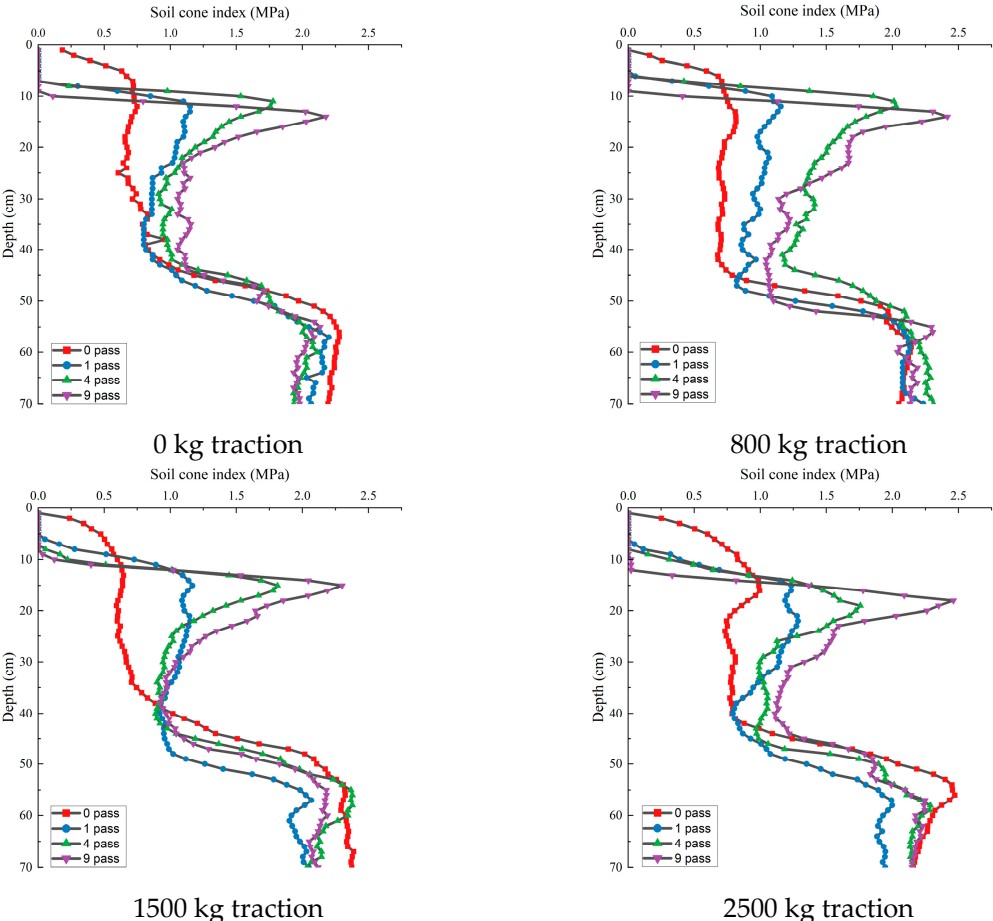

0 kg traction    800 kg traction

1500 kg traction    2500 kg traction

**Figure 14.** Soil cone index in different traction conditions with different numbers of passes.

Table 2 shows the soil cone index results focused on the nose zone area. Soil cone index results in different traction conditions and the number of passes were compared between each group.

**Table 2.** Cone index analysis of the nose zone. Lowercase letters (abc) indicate the comparison under different pass conditions within the same traction group (vertical). Capital letters indicate the comparison under different traction conditions within the same pass condition (horizontal).

| Range of the Nose Zone (cm) | | | | |
|---|---|---|---|---|
| **Pass** | **0 kg** | **800 kg** | **1500 kg** | **2500 kg** |
| 1 | 17.07bB | 20.17aA | 17.07cB | 16.23bB |
| 4 | 19.93aA | 19.2aA | 18.1bA | 19.1aA |
| 9 | 20.33aAB | 18.57aBC | 21.17aA | 17.5bC |
| **Depth of the max cone index (cm)** | | | | |
| **pass** | **0 kg** | **800 kg** | **1500 kg** | **2500 kg** |
| 1 | 12.5bC | 12.1bC | 15.03aB | 22.17aA |
| 4 | 11.33bC | 11.3bC | 15.07aB | 19.07bA |
| 9 | 14.17aB | 14.1aB | 15.07aB | 18.03bA |
| **Max cone index (MPa)** | | | | |
| **pass** | **0 kg** | **800 kg** | **1500 kg** | **2500 kg** |
| 1 | 1.15cA | 1.15cA | 1.17cA | 1.29cA |
| 4 | 1.78bA | 1.9bA | 1.81bA | 1.75bA |
| 9 | 2.17aC | 2.42aA | 2.3aB | 2.46aA |
| **Average cone index (MPa)** | | | | |
| **pass** | **0 kg** | **800 kg** | **1500 kg** | **2500 kg** |
| 1 | 0.765cB | 0.839cA | 0.838bA | 0.824cA |
| 4 | 0.931bB | 1.042bA | 0.899bB | 0.92bB |
| 9 | 0.979aC | 1.124aA | 1.010bBC | 1.075aAB |

The range of the nose zone did not change significantly with increasing passes at 0 and 800 kg traction in most situations. However, the range increases with the number of passes when the traction is under 1500 and 2500 kg conditions. For different traction conditions and the same number of passes, the range of soil cone index did not differ significantly for most of the traction conditions between one and four passes. After nine passes, there is a significant difference in the range of soil cone index between traction conditions.

For the depth of max soil cone index, the depth of max soil cone index increases with the number of passes, increasing with 0 and 800 kg traction conditions. However, under 1500 kg traction conditions, it did not change significantly with one, four and nine passes. Moreover, under 2500 kg traction conditions, the depth of the max soil cone index reduced as the number of passes increased. For different traction conditions and the same number of passes, the depth of the max soil cone index increased with the traction increasing significantly.

For the max soil cone index, the value increased with the number of passes increasing in each traction condition. No significant difference in max cone index was found for the different traction conditions at one and four passes. After nine passes, a significant difference was found at different tractions. However, the 800 kg traction had a higher max cone index than the 1500 kg traction condition.

For the average soil cone index from 0 to 40 cm, the average soil cone index increases significantly as the number of passes increases in all four traction conditions. The results after one and four passes show the soil cone index increasing significantly with the traction increase. However, the 800 kg traction condition had the highest average soil cone index value after nine passes rather than the highest traction condition, 2500 kg, as expected.

### 3.4. Profile Meter after the Compaction

The soil profiles for different traction forces are shown in Figure 15. The results show that the soil profile increases with the number of passes for all traction conditions. Although the soil profile shows a higher value after four and nine passes in higher traction conditions, there were no statistically significant differences in the soil profile value for each traction condition.

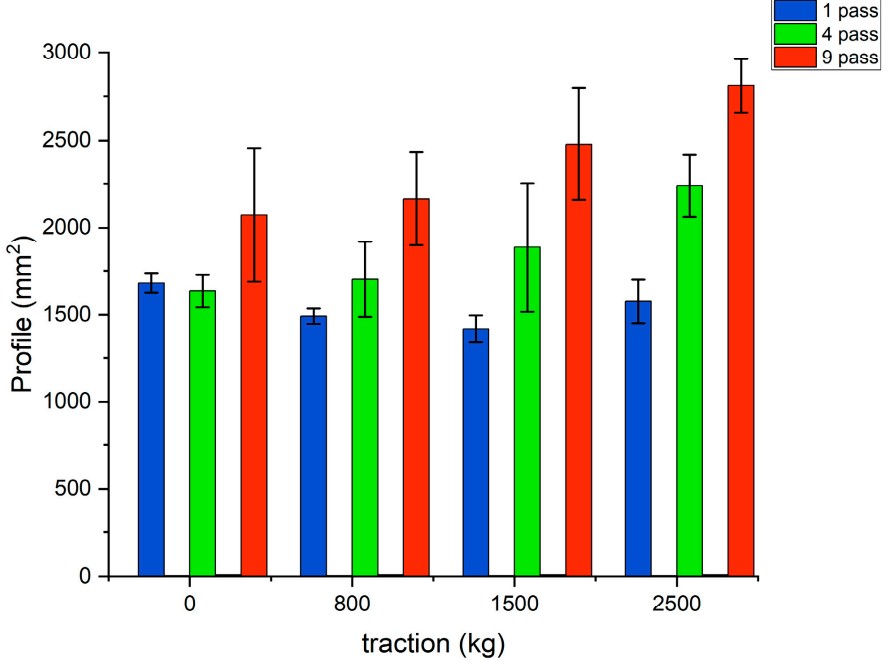

**Figure 15.** Soil profile meter after passes of different traction conditions.

### 3.5. Slip Rate

In this section, the slip rates under different soil conditions and traction conditions were analysed. The specific results are listed below.

The slip rate (%) under repeated wheeling in different traction conditions is shown in Table 3. The results show that no slip rate exists in the first pass at 0 kg. The first pass in three traction conditions had the maximum slip rate compared to the conditions of two to nine passes. No significant difference in slip rate was found between two and nine passes in all four traction conditions.

**Table 3.** Slip rate (%) under repeated wheeling.

| Pass | Traction (kg) | | | |
| --- | --- | --- | --- | --- |
| | 0 | 800 | 1500 | 2500 |
| 1 | 0.00 | 7.95 | 15.77 | 40.11 |
| 2 | 1.29 | 6.54 | 14.42 | 30.79 |
| 3 | 1.15 | 5.86 | 14.42 | 29.78 |
| 4 | 1.52 | 5.70 | 14.64 | 29.83 |
| 5 | 1.37 | 7.05 | 14.81 | 27.75 |
| 6 | 1.40 | 6.20 | 14.81 | 29.78 |
| 7 | 1.22 | 5.92 | 14.64 | 28.99 |
| 8 | 1.89 | 5.30 | 14.42 | 28.71 |
| 9 | 1.66 | 6.37 | 14.47 | 29.50 |



Rolling resistance data were collected after the number of passes, as shown in Figure 16. Trendlines were made using the method of single exponential decay. The results indicate that the first pass of the tractor has the highest rolling resistance. The rolling resistance gradually decreases to a stable area as the number of passes increases in the same traffic lane.

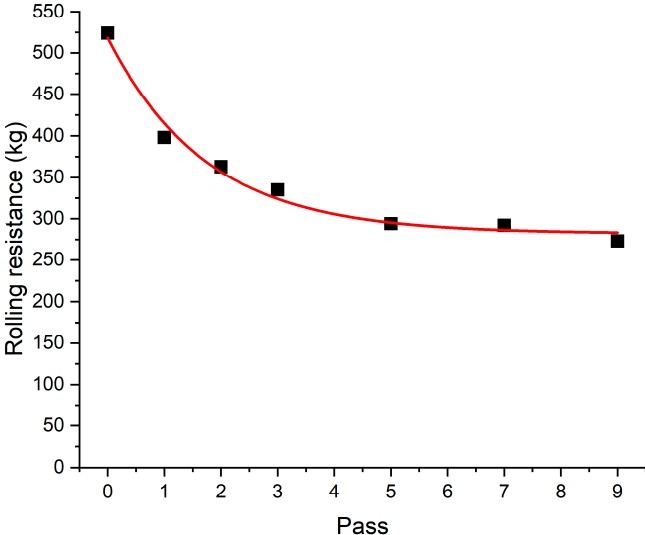

**Figure 16.** Rolling resistance collected after the number of passes.

Here is the slippage rate in different traffic and traction conditions. Three kinds of traffic conditions were chosen as the original field, which is an uncompacted field (zero passes), the trafficked field after nine compactions (nine passes) and the permanent traffic lane, which is located on the edge of the field used for transporting the machine during the farming operation. Trendlines were made using the method of single exponential decay. The results show that the slip rate increased with the traction in all three soil conditions. After nine passes, the permanent traffic lane has a lower slip rate than the zero-pass field in all traction conditions. The slip rate in the uncompacted field rises significantly with the increasing traction compared to the nine-pass field and the permanent traffic lane. Compared to Figures 16 and 17, the hard traffic lane has lower rolling resistance and slip rate, saving more energy and working efficiency than conventional agriculture.

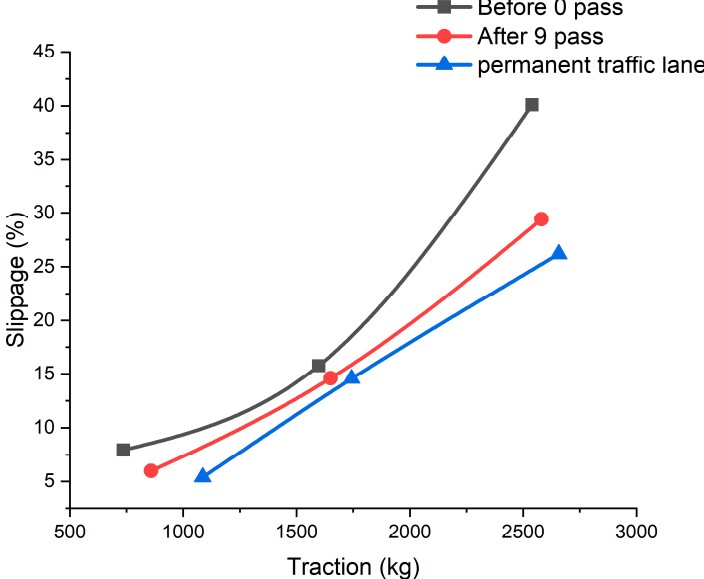

**Figure 17.** Slippage rate in different traffic conditions.

Table 4 shows the fuel consumption with different traction levels and the number of passes. The calculation of power used in Table 4 considers the net tractive effort, slip rate and energy loss in the transmission and hydraulic system. Fuel consumption is calculated by the power used. Fuel savings in each pass situation were calculated by comparing with fuel savings with zero passes in each traction condition.

**Table 4.** Fuel consumption with different traction levels and number of passes.

| Pass | Power Used (kW) | | | Fuel Consumption (kg/h) | | | Fuel Savings (%) | | |
| | 800 | 1500 | 2500 | 800 | 1500 | 2500 | 800 | 1500 | 2500 |
|---|---|---|---|---|---|---|---|---|---|
| 0 | 13.86 | 25.92 | 56.13 | 3.60 | 6.74 | 14.59 | 0.00 | 0.00 | 0.00 |
| 1 | 13.91 | 26.62 | 45.85 | 3.62 | 6.92 | 11.92 | −0.37 | −2.68 | 18.31 |
| 2 | 12.36 | 26.14 | 44.08 | 3.21 | 6.80 | 11.46 | 10.82 | −0.86 | 21.48 |
| 3 | 12.93 | 22.91 | 43.81 | 3.36 | 5.96 | 11.39 | 6.73 | 11.63 | 21.95 |
| 4 | 11.61 | 23.56 | 41.42 | 3.02 | 6.13 | 10.77 | 16.22 | 9.11 | 26.21 |
| 5 | 13.16 | 23.21 | 44.30 | 3.42 | 6.04 | 11.52 | 5.01 | 10.45 | 21.09 |
| 6 | 13.39 | 22.85 | 42.68 | 3.48 | 5.94 | 11.10 | 3.39 | 11.86 | 23.96 |
| 7 | 12.48 | 22.39 | 43.81 | 3.25 | 5.82 | 11.39 | 9.93 | 13.61 | 21.96 |
| 8 | 13.23 | 24.19 | 43.73 | 3.44 | 6.29 | 11.37 | 4.53 | 6.68 | 22.09 |
| 9 | 11.72 | 22.30 | 44.23 | 3.05 | 5.80 | 11.50 | 15.40 | 13.98 | 21.20 |
| Permanente lane | 10.32 | 18.34 | 32.37 | 2.68 | 4.77 | 8.42 | 25.50 | 29.23 | 42.34 |

As the number of passes increases, the power loss due to rolling resistance and slip decreases, and the useful power available for traction increases. As the number of passes increases, the machine uses less power for all traction levels to produce the same work. On the uncompacted field (the field without pass) and compacted field (the field after 9 pass) all traction levels require more power than the permanent traffic lane.

From an environmental point of view, fuel savings are relative to reducing $CO_2$ emissions into the atmosphere. As the number of passes increases, the machine produces less $CO_2$ for all traction levels for the same work. All traction levels produce more $CO_2$ on the uncompacted field (the field without pass) and compacted field (the field after 9 pass) than the permanent traffic lane.

## 4. Discussion

### 4.1. Effect of Traction

The stress in the soil was calculated by vertical load, horizontal load and radial normal stress at each collection point. Horizontal stress (shear stress) on the soil surface can be calculated from the given traction [85] as one part of the stress which creates the soil compaction. Our results show that the stress underneath the soil, soil bulk density, soil cone index and soil disturbance increased with the increasing traction in our field experiment. Higher traction had higher soil stress at different soil depths and locations were found in both collected and simulation results. A similar result was also found in previous research [86]. Higher pressure underneath the soil causes more compaction and increases the soil bulk density and soil cone index in each depth [75,85,87].

Many researchers have found that higher traction has more compaction [50,52,65,86]. Moreover, the static pressure distribution of the compaction procedures was studied. However, the shear stress distribution caused by the traction was not considered in soil compaction research but probably contributes significantly to soil structure deterioration [88]. Furthermore, many factors can influence the distribution of shear force in the soil under different soil conditions with different parameters. For example, tractor tyre size, pressure, the weight of the tractor, four or rear-wheel drive of the tractor, the distance between the front and rear wheels of the tractor and how the PTO is hooked up to the working part all affect the shear force caused by the traction [89]. The method of increasing the traction resistance without adding weight was implemented in the field experiment. The results confirm that higher traction increases the soil compaction [58,86,89,90] by using the brake power from the rear tractor rather than adding the machine's weight during the

test. However, the method of how the tractor is hooked up to the other, the equipment used to connect the two tractors and the stability of the tractor's operation all impact the experimental results, so further research is needed [89].

### 4.2. Effect of Repeated Wheeling

Significant differences were found in soil bulk density, soil cone index and soil profile after repeated wheeling compared to the uncompacted field. The main change in the soil bulk density and soil profile occurs in the first pass compared to the next eight passes. The soil cone index changes mainly happened from 0 to 40 cm, with the number of passes increasing. Additionally, the sinking of the topsoil was observed during repeated wheeling. The trend of the changing values in soil bulk density, soil cone index and soil profile is similar in different traction conditions. Compared to the changing results, soil moisture and stress showed no significant differences in repeated wheeling.

Other studies have also observed that the first pass forms a harder soil surface in the form of wheel ruts [91]. It also causes maximum near-surface deformation [92]. As for the topsoil's sinking, the soil layer's thickness decreased significantly from the first to the second passes, while no differences were found in the subsequent ten passes [21]. Our experiments were carried out in a dry condition. This damage to the top layer of soil due to the first and second compaction further increases the contact area between the tyre and soil, reducing the pressure per unit area caused by the tyre on the soil. Specifically, the increased contact area minimises the soil pressure per unit area. However, as the number of passes increases, the soil pressure increases due to a tighter soil structure. This phenomenon has also been observed in previous studies [86,93].

### 4.3. Effect of Slippage

The experimental results show that the slipping rate increases with the traction increasing in different soil conditions. The tractor had a lower slip rate in compacted soil where the nine passes had been completed. Furthermore, the permanent traffic lane has an even lower slip rate than the compacted field. The experimental results show that the permanent traffic lane has the lowest slip rate and highest working efficiency compared to the uncompacted and compacted fields. The limited slip rate and high working efficiency save more energy and produce less $CO_2$, guaranteeing economic and ecological benefits. Similar results were obtained in the previous study [79,94]. Higher slip causes higher soil compaction [79,95,96], which has a significant impact on soil erosion [80] and causes great damage to soil fertility [46].

Fuel economy on the permanent traffic lane may have several practical implications. First, lower fuel consumption reduces operating expenses. Economic savings can be easily obtained by multiplying the litres of fuel per hour by the price per litre purchased. According to our tests, a higher traction load has higher energy savings and efficiency, which was also obtained by the previous study [97]. Therefore, farms can use fuels in different ways. One of these is the possibility of higher quality operations with the same power used, such as better preparation of seedbeds or other tillage operations. Another example is in performing split fertilisation to improve the uptake of inputs by plants. Other possibilities include using the saved fuel for other operations such as irrigation, or the option of using the power saved by the machine to increase work capacity and time.

One strategy to reduce power loss due to skidding and rolling is to ensure traffic in the permanent traffic lane. This is one of the advantages of using a CTF system to organise the viability of field machinery [34,98,99]. For this reason, in the CTF system, the permanent traffic lane reduces the slippage within limits compared to the conventional tillage field for the same load and working resistance [11]. Furthermore, the field experiment made it possible to evaluate and quantify the machine's slip rate and rolling effects in terms of power loss and associated fuel consumption under various transport conditions. Therefore, farmers should be advised in agricultural operations to consider controlled traffic farming (CTF) systems to improve efficiency and reduce energy consumption. Despite

the difficulties of changing from conventional agriculture to CTF [13,34], planned driving trajectories for tractors rather than random movements can increase efficiency and reduce operating costs.

## 5. Conclusions

Tractor performance was measured under different traction resistance conditions in this field experiment. The study quantified the soil stress, soil bulk density, soil moisture, soil cone index, soil surface disturbance, rolling resistance and slip rate under different numbers of passes and soil and traction conditions. The actual power used under different traction conditions was collected in uncompacted soil, compacted soil and the permanent traffic lane. Fuel consumption and savings were calculated between uncompacted soil, compacted soil and the permanent traffic lane.

The results show that soil stress increases in each location as traction and the number of passes increase. Soil's physical properties increase, such as the soil bulk density, soil moisture, soil cone index and soil surface disturbance, with the increasing traction and number of passes. However, no significant difference was found between different traction conditions for the different number of passes. The slip rate increases with traction in each soil condition as uncompacted soil, compacted soil and the permanent traffic lane. The results show that the permanent traffic lane has a lower slip rate under different traction conditions than the uncompacted and compacted soil.

Furthermore, the permanent traffic lane has less energy consumption with the same traction resistance. The permanent traffic lane saved 25.50%, 29.23% and 42.34% fuel compared to the uncompacted field in 7.85, 14.71 and 24.52 kN traction conditions, respectively. Our results show that the traffic lane not only could reduce the negative effect of the soil compaction caused by the random traffic, but also could increase the working efficiency and save energy. Moreover, the dynamic factors such as traction and rolling resistance should be considered in soil compaction research rather than static weight only. In practice, the controlled traffic farming (CTF) system or driving the tractor more frequently on the permanent traffic lane should be considered to improve working efficiency and reduce energy consumption.

**Author Contributions:** Conceptualisation, L.S.; methodology, L.S. and F.G.; validation, K.L. and M.B.; formal analysis, K.L.; data curation, K.L., F.G. and M.B.; writing—original draft preparation, K.L.; writing—review and editing, M.S., M.B. and L.S.; supervision, L.S.; project administration, L.S.; funding acquisition, L.S. All authors have read and agreed to the published version of the manuscript.

**Funding:** The research activity of Kaihua Liu is financially supported by the grant from the China Scholarship Council (CSC).

**Data Availability Statement:** Data presented in this study are available on request from the corresponding author.

**Conflicts of Interest:** The authors declare no conflict of interest.

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
