# Peer review of "Soil Compaction under Different Traction Resistance Conditions—A Case Study in North Italy"

_agriculture, doi:10.3390/agriculture12111954_

Round 1
Reviewer 1 Report
Dear Authors,
Thank you for submitting your manuscript.
I have a few recommendations:
Introduction
L. 30-32 The current study shows... - There are two out of three references older than 11 years.
L. 74 CTF is a solution to soil compaction.. - Does it mean that the problem is solved and there is no place for research?
There should be one paragraph containing the aims of this study at the end of the introduction chapter.
Materials and Methods
What was the topsoil moisture?
L. 102 The organic matter content of the topsoil.. - What method did authors use?
It would be more interesting for readers if authors would change (more appealing) the names of the figures - 1,2,3,4,5,7,8,10.
I highly recommend reducing the number of figures. For example, Figure 2 is not necessary - authors can use Figure 3 for the same description. Figure 1 should be combined with Figure 4.
L. 153 Three probes were installed... How?
L. 153-154 Why did the authors use three probes into the centre of the track and only two probes into the track edge?
Figures 5 and 7 need to be improved. There should be units. Are those figures necessary? Both of them?
Figure 6 - Why do authors have negative values of the soil depth?
Double-check the spacing and subscripts in the whole document.
L. 235 - kW?
L.262-3 For example, they considered... Who "they"?
It is necessary to explain what the 20V,40V,60V,20L,40L mean.
What were the previous field management practices?
How did the authors measure the cone index into the depth of 70 centimetres? With respect to the fact that authors mentioned on line 99 - soil moisture content was dry during the experiment..
Information in tables 4 and 5 overlaps - there should be only one table.
Chapter 4.3 - There is no confrontation of obtained results with other authors.
Conclusion
Authors should point out the new findings of their study.
Author Response
Dear reviewer,
Thank you for the time and effort that put into reviewing the previous version of the manuscript. Your suggestions have enabled us to improve our work. Based on the instructions provided in your letter, we uploaded the revised manuscript file. Accordingly, we uploaded a copy of the original manuscript with all highlighted changes using the track changes mode in MS Word.
Attached to this letter is our point-by-point response to the comments raised by the reviewers.
Thank you for your help.
Kind regrads,
Kaihua Liu
Author's Reply to the Review Report (Reviewer 1)
Introduction
- 30-32 The current study shows...- There are two out of three references older than 11 years.
“current” changed into “previous”
- 74 CTF is a solution to soil compaction..- Does it mean that the problem is solved and there is no place for research?
“CTF is a solution to soil compaction” changed in to “The CTF is one of the solutions to reducing the soil compaction”
There should be one paragraph containing the aims of this study at the end of the introduction chapter.
Done
Materials and Methods
What was the topsoil moisture?
Changed” topsoil” into “topsoil (0–30 cm)”
- 102 The organic matter content of the topsoil.. - What method did authors use?
The organic matter content of the topsoil (0–30 cm) was 1.81% (Piccoli et al., 2020) refer to another field also located at experimental farm of Padova University (straight line distance not exceeding 200m between two fields).
It would be more interesting for readers if authors would change (more appealing) the names of the figures - 1,2,3,4,5,7,8,10. I highly recommend reducing the number of figures. For example, Figure 2 is not necessary - authors can use Figure 3 for the same description. Figure 1 should be combined with Figure 4.
Figure 1 add the location of the field and combined with Figure 4
Figure 2 and Figure 3 were combined
Figure 5 added the target of the Bolling probe to measure the mean normal stress
Figure 7 added the target of the soil bulk density collections in each position
Figure 8 added the explanation of the soil cone index mearsurement
Figure 10 added the explanation of how to measure the soil disturbed after diffenret number of passes
- 153 Three probes were installed... How?
Added the explanation of the installation of the Booling probe in the draft.
“For the installation of the Bolling Probe, the drill and reamer were inserted into the soil at the specific angle on the side of the probes by using a special steel frame which could ensure the angle consistence during the installation of the drill, reamer and the Bolling probe. After completion of reaming, the probe was inserted into the soil and tested for a good contact with the soil to ensure accurate data collection.”
If there is other doubt, there are some previous published paper to explain more about the Bolling probe (Berisso et al., 2013; Bolling, 1987; Keller et al., 2016; ten Damme et al., 2020).
- 153-154 Why did the authors use three probes into the centre of the track and only two probes into the track edge?
In this experiment, not only the mean normal stress of the soil in vertical direction was measured but also inlateral direction. The soil mean normal stress in the vertical direction can be applied to the depth of the subsoil (0-100cm)(Olsen, 1994). However, the lateral compaction affects shallower soil (ten Damme et al., 2020), which also verified by the simulation results in the following result. Taking into account the amount of pressure that can occur at different positions, three Bolling probe were used in the vertical direction (0-60cm) and two in the lateral position (0-40cm) in this measurement.
Figures 5 and 7 need to be improved. There should be units. Are those figures necessary? Both of them?
Unit added
Figure 6 - Why do authors have negative values of the soil depth?
We added the explanation of this problem that the figure 6 was generated by the terranimo system automaticly. https://se.terranimo.world/expert
“The simulation result and the collected data during the field experiment were stated in the result section. It is worth noting that the soil depths on the y-axis are negative numbers generated automatically by the system.
I deleted the negative “-”..
Double-check the spacing and subscripts in the whole document.
Checked
- 235 - kW?
Fixed
L.262-3 For example, they considered... Who "they"?
Fixed
It is necessary to explain what the 20V,40V,60V,20L,40L mean.
fixed
What were the previous field management practices?
Description added
How did the authors measure the cone index into the depth of 70 centimetres? With respect to the fact that authors mentioned on line 99 - soil moisture content was dry during the experiment.
A detailed explanation was added to the draft about the machine that used during the measurement of the soil cone index
As shown in the Figure, the instrument is mounted on a specially designed iron frame fixed to the hydraulic piston. The iron frame allows the penetrometer to change the location of measurement horizontally. And the hydraulic piston driven by the tractor allows the uniform insertion speed during the measurement.
Information in tables 4 and 5 overlaps - there should be only one table.
Table 5 deleted as they were saying about more or less the same thing
Chapter 4.3 - There is no confrontation of obtained results with other authors.
Fixed
Conclusion
Authors should point out the new findings of their study.
The conclusion was rewrited in a more detailed way.
Berisso, F.E., Schjønning, P., Lamandé, M., Weisskopf, P., Stettler, M., Keller, T., 2013. Effects of the stress field induced by a running tyre on the soil pore system. Soil Tillage Res. 131, 36–46. https://doi.org/10.1016/j.still.2013.03.005
Bolling, I., 1987. Bodenverdichtung und Triebkraftverhalten bei Reifen-Neue Meß-und Rechenmethoden. Lehrstuhl für Landmaschinen, Techn. Univ.
Keller, T., Ruiz, S., Stettler, M., Berli, M., 2016. Determining Soil Stress beneath a Tire: Measurements and Simulations. Soil Sci. Soc. Am. J. 80, 541–553. https://doi.org/10.2136/sssaj2015.07.0252
Keller, T., Trautner, A., Arvidsson, J., 2002. Stress distribution and soil displacement under a rubber-tracked and a wheeled tractor during ploughing, both on-land and within furrows. Soil Tillage Res. 68, 39–47. https://doi.org/10.1016/S0167-1987(02)00082-X
Olsen, H.J., 1994. Calculation of subsoil stresses. Soil Tillage Res. 29, 111–123. https://doi.org/10.1016/0167-1987(94)90047-7
Piccoli, I., Sartori, F., Polese, R., Berti, A., 2020. Crop yield after 5 decades of contrasting residue management. Nutr. Cycl. Agroecosystems 117, 231–241. https://doi.org/10.1007/s10705-020-10067-9
ten Damme, L., Schjønning, P., J. Munkholm, L., Green, O., K. Nielsen, S., Lamandé, M., 2021. Soil structure response to field traffic: Effects of traction and repeated wheeling. Soil Tillage Res. 213. https://doi.org/10.1016/j.still.2021.105128
ten Damme, L., Stettler, M., Pinet, F., Vervaet, P., Keller, T., Munkholm, L.J., Lamandé, M., 2020. Construction of modern wide, low-inflation pressure tyres per se does not affect soil stress. Soil Tillage Res. 204, 104708. https://doi.org/10.1016/j.still.2020.104708

Reviewer 2 Report
I think the article entitled "Soil compaction under different traction resistance conditions- A case study in North Italy" is interesting, especially for agricultural practice, and could be published in the journal Agriculture.
The authors obtained data in a field experiment. The data is up-to-date, from 2022. The results of the field research were correctly obtained, compiled and presented. However, the paper requires improvement, corrections and additions. Detailed comments and suggestions are shown in the manuscript as comments.

Author Response
Dear reviewer,
Thank you for the time and effort that they have put into reviewing the previous version of the manuscript. Their suggestions have enabled us to improve our work. Based on the instructions provided in your letter, we uploaded the file of the revised manuscript. Accordingly, we have uploaded a copy of the original manuscript with all the changes highlighted by using the track changes mode in MS Word.
Attached to this letter is our point-by-point response to the comments raised by the reviewers.
Thank you for your help.
Kind regrads,
Kaihua Liu

Reviewer 3 Report
In the entire text, words and signs are sometimes joined, please separate them, for example line 33 - operations(8-12) or line 52 presure (29-31).
Sometimes you write, for example, row 97 - 850 mm, so that in row 98 you would write 50-60 mm. Please make it even
Row 138 Figure3
Row 181 you start the sentence with a small letter - soil bulk density
In the text below the pictures, somewhere there is a dot at the end, somewhere not
Row 211 ...9 passes. - bring the point closer
Row 250 CO2 is written incorrectly
It is MANDATORY to state and explain the applied statistics at the end of the materials and methods
In Figure 13, indicate the units for soil moisture
Figure 14, how did you express the density of the soil in g/cm2 - Isn't the density in g/cm3? Please explain
I am confused by the conclusion on 318-322, please clarify
Red 335 hardpan is artificial or nature soil layer
Row 395 and 414 start the title in table 3 and figure 19 with a lowercase letter
Row 465 What is PTO?
In Table 5 CO2 is misspelled, also in lines 441-446 CO2 is misspelled
Author Response
Dear reviewer,
Thank you for the time and effort that they have put into reviewing the previous version of the manuscript. Their suggestions have enabled us to improve our work. Based on the instructions provided in your letter, we uploaded the file of the revised manuscript. Accordingly, we have uploaded a copy of the original manuscript with all the changes highlighted by using the track changes mode in MS Word.
Attached to this letter is our point-by-point response to the comments raised by the reviewers.
Thank you for your help.
Kind regrads,
Kaihua Liu
Author's Reply to the Review Report (Reviewer 3)
In the entire text, words and signs are sometimes joined. Please separate them, for example, line 33 - operations(8-12) or line 52 pressure (29-31).
Fixed
Sometimes you write, for example, row 97 - 850 mm, so that in row 98 you would write 50-60 mm. Please make it even
Fixed
Row 138 Figure3
Fixed with the correct space
Row 181 you start the sentence with a small letter - soil bulk density
Fixed
In the text below the pictures, somewhere there is a dot at the end, somewhere not
Fixed
Row 211 ...9 passes. - bring the point closer
Fixed
Row 250 CO2 is written incorrectly
Fixed the mistake of the CO2 and delete the repeat part of the result
It is MANDATORY to state and explain the applied statistics at the end of the materials and methods
Added the statistics method in the result part
In Figure 13, indicate the units for soil moisture
Fixed, changed into “ %” now.
Figure 14, how did you express the density of the soil in g/cm2 - Isn't the density in g/cm3? Please explain
Sorry, the mistake fixed now.
I am confused by the conclusion on 318-322, please clarify
Deleted that part.
Red 335 hardpan is artificial or nature soil layer
Added the explanation of the hardpan formation caused by the repeat deep plowing during these years of the field
Row 395 and 414 start the title in table 3 and figure 19 with a lowercase letter
Fixed
Row 465 What is PTO?
PTO is tractor power take-off unit, which has added the explaination in the paper.
In Table 5 CO2 is misspelt, also in lines, 441-446 CO2 is misspelt
Sorry, the mistake has been fixed.

Round 2
Reviewer 1 Report
Dear authors,
Thank you for your corrections.
I have a few reccomendations:
What was the topsoil moisture? I would like to see the topsoil moisture value in the text..
L. 369-370 - The analysis of variance (ANOVA) was used to compare means with a probability level of 5%. Did the data have a normal distribution?
L. 421-423 There are five groups of 421 bars as 20 cm depth in vertical (20V), 40 cm depth in vertical (40V), 60 cm depth in vertical 422 (60V), 20 cm depth in lateral (20L) and 40 cm depth in lateral (40L). - That should be in chapter - Materials and methods.
L.699, 701 Author mentioned - "different soil" what does it mean? Different types (probably not) - should be changed.
Author Response
Dear reviewer,
Based on the instructions provided in your letter, we uploaded the revised manuscript file. Accordingly, we uploaded a copy of the original manuscript with all highlighted changes this time.
Attached to this letter is our point-by-point response to the comments raised by the reviewers.
Thank you for your help.
Kind regrads,
Kaihua Liu
